# First Molecular Detection of Zoonotic *Chlamydia* Species in Vietnamese Goats

**DOI:** 10.3390/pathogens11080903

**Published:** 2022-08-10

**Authors:** Valentina Chisu, Rosanna Zobba, Giovanna Masala, Thanh Loan Tran, Quynh Tram Ngo Viet, Dinh Binh Tran, Hoang Bach Nguyen, Khanh Toan Tran, Alberto Alberti

**Affiliations:** 1Department of Animal Health, Istituto Zooprofilattico Sperimentale della Sardegna, Via Vienna, 2, 07100 Sassari, Italy; 2Department of Veterinary Medicine, University of Sassari, Via Vienna 2, 07100 Sassari, Italy; 3Department of Immunology and Pathophysiology, Hue University of Medicine and Pharmacy, Hue University, Hue 530000, Vietnam; 4Department of Microbiology, Hue University of Medicine and Pharmacy, Hue University, Hue 530000, Vietnam; 5Department of Testing and Quality Assurance, Hue University of Medicine and Pharmacy, Hue University, Hue 530000, Vietnam

**Keywords:** zoonoses, Vietnamese goats, *Chlamydia abortus*, *Chlamydia psittaci*

## Abstract

The genus *Chlamydia* comprises obligate intracellular bacteria that infect a wide variety of hosts, with infection leading to a range of diseases in humans and animals; they thus constitute a major public health threat. Among the members of the *Chlamydiaceae* family, *Chlamydia suis, C. abortus, C. pecorum*, and *C. psittaci* represent the most important pathogenic species infecting a large range of hosts and are a well-established threat to livestock. Information regarding the circulation of *Chlamydia* species in ruminants from Vietnam is lacking. In this study, DNA extracted from 60 blood samples collected from goats in Hue province was used for *Chlamydia* spp. identification by classic PCR and Sanger sequencing. *Chlamydia* spp. were detected in eleven samples (18.3%) and *C. abortus* and *C. psittaci* were molecularly identified by sequencing. Despite the limited sample size in this study, findings point out the relevance of ruminants as hosts of chlamydial species in Central Vietnam and the importance of monitoring chlamydial strains through the activation of surveillance programs in this country. The need for a deeper evaluation of human and animal health risk analysis in terms of chlamydiosis should be also considered.

## 1. Introduction

The order *Chlamydiales* includes different chlamydial species with an obligate intracellular nature and capable of replicating within eukaryotic cells of different origin. Two forms of the bacterium can be observed, corresponding to a biphasic developmental cycle. The elementary body (EB) of the microorganism is the replicating form, whereas the reticulate body (RB) corresponds to the non-replicating infectious particle that is released upon rupture of infected cells [1,2]. The taxonomy of the genus *Chlamydia* has undergone continuous revision, and, to date, the order *Chlamydiales* is composed of nine families, with novel species frequently described on the basis of a combination of biological and genetic data [3]. The potential host range of bacteria in this order includes over 400 globally documented species, including humans, domestic and wild vertebrates, reptiles, amphibians, fish and amoebae [4]. Livestock animals, especially ruminants, and birds are the most frequent reservoirs leading to human transmission [5,6]. Cases of infection have been recently reported among wild ungulates and marsupials [7].

At present, members belonging to the family *Chlamydiaceae* are the best-described species of this order associated with disease in humans and animals. *Chlamydia psittaci*, *C. abortus*, *C. caviae*, *C. felis*, and *C. suis* are frequently reported as zoonoses transmitted to humans from infected animals (birds or livestock) [8]. However, interest in other families within this order and collectively called *Chlamydia*-related bacteria (CRB) [9,10,11] is also rising, as they constitute a source of infection for humans and animals. Infections related to bacteria belonging to the *Parachlamydiaceae* family have been increasingly reported over the past few decades, and some reports support their role as agents of community-acquired pneumonia (CAP) in humans, and as a cause of infertility, abortion and respiratory infections in ruminants [12].

The detection and control of chlamydial infections are important issues, and the identification and clinical diagnosis of chlamydial agents are still challenging and hampered by the absence of specific clinical signs. Indeed, clinical symptoms of chlamydiosis overlap with symptoms related to other tick-borne diseases related to distinct etiological agents (e.g., *Rickettsia*, *Anaplasma*, *Coxiella*), and can range from asymptomatic infection to mild fever or severe disease with life-threatening illnesses in immunocompetent individuals. Various disease manifestations, such as conjunctivitis, rhinitis, pneumonia, mastitis, placentitis leading to abortion, stillbirth or weak neonates, infertility and enteritis, have been reported in chlamydial-infected hosts [13]. Since serological studies to determine the status of chlamydiosis in livestock have revealed the prevalence of this disease to a variable extent in both diseased and normal animals [6], the clinical picture of infection depends on both the virulence of the infecting strain of *Chlamydia* and on specific risk factors in the infected animal. In addition, the management of *Chlamydial* infection depends on the clinical presentation of the disease, and the difficulty in the diagnosis of the bacterium could invalidate the real prevalence of animal chlamydiosis in a specific geographical area. In Vietnam, the number of sheep and goats has risen rapidly, increasing from almost 1.5 million animals in 2006 to 4,170,000 in 2015 (https://www.fao.org/ppr/current-situation/country-detail/en/?country_iso3=VNM accessed on 3 May 2022). Chlamydial diseases amongst ruminants are currently unknown in Vietnam, as well the global impact of *Chlamydia* on public health, which has largely been underestimated. However, several cases of community-acquired pneumonia (CAP) caused by *Chlamydia psittaci* were reported in 2012 in Vietnamese patients with unknown exposure history to chlamydial reservoir animals [14].

Since goats play a significant role in the transmission of the disease to humans and domestic ruminant farming is common in Vietnam, the aim of this preliminary study was to investigate the presence of *Chlamydia* 16SrRNA species in goats sampled in six small rural communities of Central Vietnam.

## 2. Results

### 2.1. Chlamydia spp. Detection

Overall, 11/60 (18%) samples collected from Vietnamese goats belonging to herds 1, 3, 4 and 5 tested positive for *Chlamydiales* DNA using 16S rRNA PCR. In order to molecularly identify *Chlamydiales* detected in ruminants in this study, the PCR products of the 11 PCR-positive samples were directly sequenced. A total of 4/11 (36.3%) samples, all belonging to the same herd, generated a clear sequencing signal and enabled BLAST search analyses against all available *Chlamydiales* 16S rRNA gene sequences from GenBank. Chromatograms resulting from the remaining PCR-positive samples were unreadable. Sequencing of the 16S rRNA PCR products and sequence comparisons resulted in the establishment of two 16S rRNA sequence types, as shown in Table 1.

### 2.2. Sequence Analyses of the 16S rRNA-Positive Samples

Based on BLASTn and subsequent phylogenetic analysis, the 16S rRNA Chlamydiales genotypes identified in this study were associated with the *Chlamydiaceae* family. Upon BLASTn comparison, the 16S rRNA sequence types were mostly similar (99–100%) to chlamydial strains recorded in GenBank as *C. abortus* and *C. psittaci*. In particular, two identical sequences identified as Genotype 1 were 100% identical to *Chlamydia psittaci* accession numbers CP047319 and CP002744, while the BLASTn analysis of the 16S rRNA target gene of the Genotype 2 sequences showed 100% identity with *C. abortus* strains isolated from various hosts worldwide (Accession Number: EF486854).

## 3. Discussion

Species belonging to the *Chlamydiaceae* family are frequently found in domestic ruminants [15], which are important reservoirs and could act as a source of new infections. Due to the potential of domestic ruminants to act as reservoirs of *Chlamydia* species, they represent also a solid threat to human infection. For this reason, the detection and control of infected herds are important issues for the development of quantitative risk assessments and appropriate control measures in veterinary and public health [16]. Intensive goat farming is not used in Vietnam as other meat sources, such as pig or chicken, are commonly exploited. However, goat farming represents in Vietnam a valuable resource to supplement household income for farmers with semi-intensive farms, usually family-run, concentrated mainly in the northern part of the Red River Delta and in some areas along the central coast. This study reports the first detection of *C. abortus* and *C. psittaci* in goats from Hue, Vietnam. In Hue, goats are farmed in small groups kept by families for meat, often in rural mountain communities. Data on the occurrence of chlamydial infections in ruminants from Southeast Asia are very limited and fragmented. Cases of *Chlamydiosis* have been described in small ruminants in China [17]. Similarly, *C. abortus* has been recently detected by PCR and sequencing in goats from India [18], allowing confirmation that goat herds could act as a source of infection. However, to the best of our knowledge, the prevalence of *Chlamydia* spp. in ruminants has never been reported in Vietnam, and, considering the high number of goats positive for *Chlamydia* spp. farmed in the six local human communities, we speculate that the impact of the disease on public health is not to be underestimated. In this study, the 11/60 samples that tested positive for *Chlamydia* species were detected using the PCR method by amplifying highly conserved regions of the chlamydial intergenic space between the 16S gene. These results were in line with previous studies that have already shown that even if the isolation of the pathogens from aborted samples represents the gold standard for definitive diagnosis, detection of *Chlamydia* specific DNAs by PCR is sensitive and highly specific [19]. This method was chosen as a diagnostic method and used to identify *Chlamydiales* in samples from this study.

Goats analyzed in this research were asymptomatic, and this confirms that the identification of infected animals is challenging because animals can be infected without being symptomatic. Infected animals could show a wide range of clinical signs or no signs of infection, and clinical manifestations of chlamydiosis are highly dependent on the metabolic status of the host species infected [13].

From the total of 11 positive samples detected, only four were sequenced. This could be due to co-infection with two or more species of *Chlamydia*, which could have affected amplicons in direct Sanger sequencing. According to this, *C. abortus* and *C. psittaci* were sequenced since they were unique genotypes present in the samples. The detection of more than one species of *Chlamydia* in the same host should be taken into account since the presence of co-infection can lead to neglecting the persistent focus of infection and a lack of disease surveillance systems. Further studies are needed to identify chlamydial co-infections in ruminants from Vietnam. From the total number of positive samples, two were identified as *C. abortus* after 16S rRNA sequencing. *C. abortus* infection, mainly associated with enzootic abortion in ewes, contributes to impacts on animal health and to a reduction in the productivity and reproductive performance of livestock, which can result in weight loss in the animal, animal welfare issues, and economic losses for the farmer [13]. The zoonotic potential is also known, and the agent can affect pregnant women who have been in contact with infected animals [20]. Thus, farmers, veterinarians, and other professionals who come into contact with *Chlamydia* should be alert when managing vaginal discharges, placentas from aborting females, or aborted fetuses because of the risk of *C. abortus* aerosol inhalation. Moreover, the role of *C. abortus* as a new, emerging, atypical pneumonia agent has been also suggested [21], indicating that *C. abortus* infection should be taken into account for differential diagnoses in respiratory patients in particular if they have had contact with aborted ruminants or their afterbirth. In our study, two samples were also identified as *C. psittaci* after sequencing. Although *C. psittaci* has been widely identified in 471 different avian species, representing the first route of infection to humans, the bacterium has also been isolated from an increasing number of mammalian hosts [22], such as ruminants (including goats, and, to a lesser degree, cattle, horses, pigs and deer), which are susceptible to *C. psittaci* infection and which act as potential reservoirs and amplifiers of *C. psittaci* infection. Among human clinical manifestations, *C. psittaci* infection can be associated with severe flu-like infections and largely non-specific features [22]. Atypical pneumonia has been commonly reported in human patients infected with *C. psittaci*. Human psittacosis was detected for the first time in Vietnam [14]. In this study, detection of *C. psittaci* from two blood samples of ruminants suggests that chlamydiosis occurrence is underestimated and that the true incidence in the human population is actually unknown.

The limited number of goats in Vietnam, usually reared in small groups by families, together with their not easily reachable locations, are the main problems related to sampling large subject numbers. Therefore, this paper represents a first effort to investigate the presence of different chlamydial species in Vietnamese farm animals. More sampling with larger animal numbers and additional farm species is needed to establish the impact of chlamydiosis in Vietnam and to develop control strategies under a One Health perspective.

## 4. Materials and Methods

### 4.1. Study Area and Sample Collection

As part of a project carried out by the Hue University of Medicine and Pharmacy, with the aim of filling the gaps in knowledge on tick-borne infections in livestock in Vietnam, samples previously collected and tested for the presence of bacterial and protozoal agents (data not published) were also tested for the presence of Chlamydiales. Specifically, 60 goats were randomly selected from six small herds (up to 30 heads) farmed in the lowland and mountainous areas of Luoi and Nam Dong districts, Thua Thien Hue province (Central Vietnam, Latitude: 16°27′42″ N Longitude: 107°35′43″ E); they were chosen at convenience during April 2018. This area’s climate is tropical, with the wet season being warm and overcast, and the dry season being hot and mostly cloudy. Over the course of the year, the temperature typically varies from 17 °C to 35 °C.

Ten goats per farm were casually collected. No avian species or other ruminants were farmed in the six properties.

All goats were adults (over 6 months) with average weight of around 15 kg. There was no previous history of abortion in the herds, even if other reproductive disorders such as stillbirth elicited the suspicion of chlamydiosis in the farms. Goats had no observable diseases or injuries. Moreover, sampled animals were all treated for ticks at the beginning of tick season. As goats in Vietnam are farmed for meat in small groups by families, the consent of the owners of the goats was obtained before collection of blood samples used for DNA extraction and detection of the presence of *Chlamydia* spp. in the samples.

### 4.2. Chlamydiales PCR Amplification, Purification, and Sequencing

The study was conducted by sampling EDTA blood samples from 60 goats. Blood was centrifugated for 10 min at 1500× *g*, and the buffy coat was taken from the top of the red cell pellet and stored at −80 °C until further analysis. DNA was extracted from all samples by using the DNeasy blood and tissue kit (QIAGEN, Chatsworth, CA, USA), according to the vendor’s recommendations for blood extraction. DNA was then stored at −20 °C until use in PCR amplification assays. The DNA samples were then delivered to the Serology Laboratory of the Istituto Zooprofilattico Sperimentale Sardegna in Sassari, Italy.

All DNA samples were amplified by conventional PCR using a pair of oligonucleotide primers (16Sfor2, 5′-CGTGGATGAGGCATGCAAGTCGA-3′ and 16Srev6, 5′-ATCTCTCAATCCGCCTAGACGTCA-3′) that produce a 298-bp fragment of the *Chlamydiales* 16S rRNA gene [9]. One microliter of the extracted DNA was added to a final volume of 25 μL of reaction mixture containing 12.5 microliters of master mix 2× (Ampliquen, Odense, Denmark), 1× PCR buffer, 200 μM of deoxynucleoside triphosphates (dNTPs), 2 mM MgCl_2_, 0.5 U of *Taq* polymerase, and 0.5 μM of each primer. PCR reactions were performed in an Eppendorf thermocycler (Eppendorf, Hamburg, Germany). Initial denaturation for 5 min at 95 °C was followed by 39 cycles of 60 s at 94 °C, 45 s at 60 °C, and 45 s at 72 °C, and a final extension step of 5 min at 72 °C. A negative control using no template and dH_2_O and a positive control using *C. abortus* DNA were included in each amplification test. The amplification results were visualized by 1.5% agarose gel electrophoresis stained with SYBR Safe DNA Gel Stain (Invitrogen) and examined under UV transillumination.

The forward and reverse chromatograms generated with 16S rRNA primers were displayed with Chromas 2.2 software (Technelysium, Helensvale, Australia). Sequences were then aligned with CLUSTALX [23], and matched against the GenBank database with nucleotide blast BLASTn [24]. The CLUSTALW program [25] was used to calculate pairwise/multiple sequence alignments and sequence similarities, while the Bioedit platform [26] was used to generate the sequence identity matrices.

## 5. Conclusions

Although the number of samples was limited, *C. abortus* and *C. psittaci* species were detected in ruminants for the first time in Vietnam. Clinicians should be informed that zoonotic *Chlamydiaceae* are circulating in this region. Further clinical studies are needed to better understand the prevalence of chlamydial agents in small ruminants and the significant role of goats in the transmission of the disease to humans.

## Figures and Tables

**Table 1 pathogens-11-00903-t001:** BLASTn identities of the 16S rRNA sequence types obtained from the eleven infected goats clustered by farm.

	Infected Goats	Farms with at Least One Infected Goat	16S rRNA PCR Positive	BLAST Analyses	Sequence Types
1	Goat 7	1	+	-	-
2	Goat 8	1	+	-	-
3	Goat 34	3	+	-	-
4	Goat 36	3	+	-	-
5	Goat 41	4	+	-	-
6	Goat 53	5	+	*C. psittaci*	Genotype1
7	Goat 55	5	+	*-*	-
8	Goat 56	5	+	*-*	-
9	Goat 57	5	+	*C. abortus*	Genotype 2
10	Goat 58	5	+	*C. psittaci*	Genotype 1
11	Goat 59	5	+	*C. abortus*	Genotype 2

## Data Availability

Not applicable.

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
