# Peer review of "First Molecular Detection of Zoonotic Chlamydia Species in Vietnamese Goats"

_pathogens, 2022, doi:10.3390/pathogens11080903_

Round 1
Reviewer 1 Report
The key omission is lack of description of sampling frame / sample sizes & why these animals were selected. "60 blood samples collected from goats in Hue province". male/female? Same/multiple herds? While all were asymptomatic there is no information on previous abortions in the herds which is important.
Without this information it is difficult to provide an overall review - it is surely relevant as novel and potential public health issue with addition of selection
Also minimal detail on study limitations aside from "limited sample size" in abstract and in the conclusion which could be expanded.
Author Response
The key omission is lack of description of sampling frame / sample sizes & why these animals were selected. "60 blood samples collected from goats in Hue province". male/female? Same/multiple herds? While all were asymptomatic there is no information on previous abortions in the herds which is important. Without this information it is difficult to provide an overall review - it is surely relevant as novel and potential public health issue with addition of selection. Also, minimal detail on study limitations aside from "limited sample size" in abstract and in the conclusion which could be expanded.
Response: Goats are not farmed intensely in Vietnam, but they are kept by families in small groups for meat. Actually, abortion was not reported in the animals, however Chlamydiosis was suspected in human living in close contact with animals. Also, sampling goats in Vietnam is not easy since getting access to rural communities is very difficult, and we managed to get some samples from different goat groups (50% male, 50% females). All this information is now incorporated in the manuscript. Finally, a section on biases and limitations of the study has been added to the discussion.
Reviewer 2 Report
The article entitled "First molecular detection of zoonotic Chlamydia species in Vietnamese
goats" by Chisu et al report first molecular detection of Chlamydia species in Vietnamese
goats. Sixty blood samples were collected from goats in Hue province and PCR was used for identification. Chlamydia spp. were detected in 11 samples (18.3%) and C. abortus and C. psittaci were molecularly identified. The article is of interest due to zoonotic nature of Chlamydia spp, but lack important information related to study design and sampling.
General comments:
It will be good to give the reader some basic information about goat farms in Hue, how many flocks are there? what are the basic management system used?
Abstract:
Line 16-18: Please, modify as:
The genus Chlamydia comprises obligate intracellular bacteria that infect a wide variety of hosts, with infection leading to a range of diseases in humans and animals, which constitute a major public health threat.
Lines 20-21: livestock already include domestic animals, please modify as:
a large range of hosts and are a well-established threat to livestock.
Line 24: obtained, please replace with detected
Line 28: should be also considered
Introduction
Lines 74-76: The results provide evidence on the presence of Chlamydia in goats but nothing about distribution, and prevalence.
Also, what do the authors mean by ruminants being relevant as a host for Chlamydial spp.? This goal made no sense because it is well known that ruminants are a primary host for a variety of Chlamydial spp. Please modify your objectives to reflect what have been reported.
Results:
Line 78:
2.1. Chlamydia spp. prevalences : please use detection instead of prevalence
When using prevalence, authors should exercise caution because information about the study design, sample collection, sample size calculation, and sampled population should be provided.
Line 91, 92: delete
Line 94: The authors didn't provide figure 1 as indicated in line 94
Discussion section:
It will be useful to add section on the biases and limitations of the study.
4. Materials and Methods
4.1. Ethical approval, study area and sample collection
Lines 162-163: Ethical approval is listed at the end of the article, so remove from here.
study area and sample collection:
This section lack many details that could have direct effects on the study results:
What the target population?
How sample size was determined? why 60 samples not 10 or 100 or 1000
How the 60 sampled goats were selected?
Were samples collected from one farm or multiple farms? how these farms were selected?
How do we know goats selected for the study were representative for goats in the target population?
4.2. Chlamydiales PCR amplification, purification and sequencing
4.2 & 4.3 exactly same title
Author Response
The article entitled "First molecular detection of zoonotic Chlamydia species in Vietnamese
goats" by Chisu et al report first molecular detection of Chlamydia species in Vietnamese
goats. Sixty blood samples were collected from goats in Hue province and PCR was used for identification. Chlamydia spp. were detected in 11 samples (18.3%) and C. abortus and C. psittaci were molecularly identified. The article is of interest due to zoonotic nature of Chlamydia spp, but lack important information related to study design and sampling.
General comments:
It will be good to give the reader some basic information about goat farms in Hue, how many flocks are there? what are the basic management system used?
Response: Information on goat farming Vietnam and details on sampling are now incorporated in the manuscript (e.g. number of goat groups, female/male ratio, and farming in Vietnam)
Abstract:
Line 16-18: Please, modify as:
The genus Chlamydia comprises obligate intracellular bacteria that infect a wide variety of hosts, with infection leading to a range of diseases in humans and animals, which constitute a major public health threat.
Response: Done
Lines 20-21: livestock already include domestic animals, please modify as:
a large range of hosts and are a well-established threat to livestock.
Response: Done
Line 24: obtained, please replace with detected
Response: Done
Line 28: should be also considered
Response: Done
Introduction
Lines 74-76: The results provide evidence on the presence of Chlamydia in goats but nothing about distribution, and prevalence.
Response: Done
Also, what do the authors mean by ruminants being relevant as a host for Chlamydial spp.? This goal made no sense because it is well known that ruminants are a primary host for a variety of Chlamydial spp. Please modify your objectives to reflect what have been reported.
Response: The last section of introduction was modified according to the reviewer comment.
Results:
Line 78:
2.1. Chlamydia spp. prevalences: please use detection instead of prevalence
When using prevalence, authors should exercise caution because information about the study design, sample collection, sample size calculation, and sampled population should be provided.
Response: Done
Line 91, 92: delete
Response: Done
Line 94: The authors didn't provide figure 1 as indicated in line 94
Response: Done
Discussion section:
It will be useful to add section on the biases and limitations of the study.
Response: A section on biases and limitations of the study has been added to the discussion
- Materials and Methods
4.1. Ethical approval, study area and sample collection
Lines 162-163: Ethical approval is listed at the end of the article, so remove from here.
study area and sample collection:
Response: Done
This section lack many details that could have direct effects on the study results:
What the target population?
How sample size was determined? why 60 samples not 10 or 100 or 1000
How the 60 sampled goats were selected?
Response: See above
Were samples collected from one farm or multiple farms? how these farms were selected?
How do we know goats selected for the study were representative for goats in the target population?
Response: See above
4.2. Chlamydiales PCR amplification, purification and sequencing
4.2 & 4.3 exactly same title
Response: Done
Round 2
Reviewer 1 Report
As stated in the original reciew the confirmation of chlamydia in goats in VietNam is significant - finding 11 positive in only 60 samples is concerning yet it is not clear how the goats are distributed. I am yet to understand the rationale for this piece of research - was its primary aim to detect chlamydia in goats following human infection with goat keeping being a risk factor? Had samples been taken for another primary reason and were then processed to detect chlamyida? It doesnt seem this was done due to abortions in goats (or other ruminants kept on farms)
The additional information about sample frame is not so helpful (and confused by the English: New paragraph line 158-165):
Goats were randomly selected from different small familiar [family?] groups. Different farms ? Different groups in same farms
"different" = how many? More than one goat per flock (to know if positives clustered by flock/group - suspecting goat 66 and 67 could be from same flock?)
Are they any linkages by value chain - source of animals - assuming not imported but bought at same market?? etc
For C.psittaci - avian species farmed on same property
For C.abortus - other ruminants farmed on same property
There was not previous history of abortion, even if chlamydiosis was suspected in the rural communities [need to clarify - does this mean in humans??].
Reference 22 : human chlamydia first detected in 2013 in VietNam - thus not recently
Though of secondary importance English needs revised throughout - with spelling eg "Richettsia" (L58)
Author Response
Thank you for your comments. The paper was improved by adding more information as requested. First, the rationale of the research is now clearer and the primary reason regarding the collection of the samples in goats has been added, as follows:
“As part of a project carried out by the Hue University of Medicine and Pharmacy with the aim of filling the gaps in knowledge on tick-borne infections in livestock in Vietnam, samples previously collected and tested for the presence of bacterial and protozoal agents (data not published) were also tested for the presence of Chlamydiales.”
The additional information about sample frame is not so helpful (and confused by the English: New paragraph line 158-165):
Response: We agree, the sentence lacked clarity and therefore has been modified as follows:
“The limited number of goats in Vietnam, usually reared in small groups by families, together with their not easily reachable locations are the main problems related to sampling large subjects’ numbers. Therefore, this paper represents a first effort to investigate the presence of different chlamydial species in Vietnamese farm animals. More sampling with larger animal numbers and additional farm species is needed to establish the impact of chlamydiosis in Vietnam and to develop control strategies under a One Health perspective.”
Goats were randomly selected from different small familiar [family?] groups. Different farms ? Different groups in same farms
"different" = how many?
Response: The number of flocks (six) from which goats were collected has been clarified in the paper that now reads:
“Specifically, 60 goats were randomly selected from six family owned small herds (up to 15-20 heads ). Ten goats per farm were casually collected.
‘Specifically, 60 goats from six small herds (up to 30 heads) farmed in the lowland and mountainous areas of Luoi and Nam Dong districts, Thua Thien Hue province (Central Vietnam, Latitude: 16°27′42″ N Longitude: 107°35′43″ E) were choosen at convenience during April 2018. This area climate is tropical, with the wet season warm and overcast, and dry season hot and mostly cloudy. Over the course of the year, the temperature typically varies from 17°C to 35°C.
Ten goats per farm were casually collected. No avian species or other ruminants were farmed in the six properties.
All goats were adult (over 6 months) with average weight of about 15 kg. There was not previous history of abortion in the herds, even if chlamydiosis was suspected in the rural communities. Goats had no observable diseases or injuries. Moreover, sampled animals were all treated against ticks at the beginning of tick season. As goats in Vietnam are farmed for meat in small groups by families, a consent from the owners of the goats was taken before collection of blood samples used for DNA extraction and detection of the presence of Chlamydia spp. in the samples.”
More than one goat per flock (to know if positives clustered by flock/group - suspecting goat 66 and 67 could be from same flock?)
Response: In Table 1 we now provided information on flocks from which goats were collected
Are they any linkages by value chain - source of animals - assuming not imported but bought at same market?? etc
For C.psittaci - avian species farmed on same property
For C.abortus - other ruminants farmed on same property
Response: This information has been added in the paper that now reads:
“No avian species or other ruminants were farmed in the six properties.”
There was not previous history of abortion, even if chlamydiosis was suspected in the rural communities [need to clarify - does this mean in humans??].
Response: The sentence has been reformulated as follows:
“There was not previous history of abortion in the herds, even if other reproductive disorders such as stillbirth elicited the suspicion of chlamydiosis in the farms.”
Reference 22 : human chlamydia first detected in 2013 in VietNam - thus not recently
Response: Modified
Though of secondary importance English needs revised throughout - with spelling eg "Richettsia" (L58)
Response: The English has been revised throughout the paper
Author Response
Thank you for your comments. Your suggestions have been added in the new version of the paper that now reads:
“ Specifically, 60 goats were randomly selected from six small herds (up to 30 heads) farmed in the lowland and mountainous areas of Luoi and Nam Dong districts, Thua Thien Hue province (Central Vietnam, Latitude: 16°27′42″ N Longitude: 107°35′43″ E) were choosen at convenience during April 2018. This area climate is tropical, with the wet season warm and overcast, and dry season hot and mostly cloudy. Over the course of the year, the temperature typically varies from 17°C to 35°C.
Ten goats per farm were casually collected. No avian species or other ruminants were farmed in the six properties.’’
If this is the first study on Chlaymedia, why the authors didn't screen larger sample of the population
with more cheaper test like ELISA?
Response: We believe this is a preliminary report in which we provided data on 16S rRNA for the first time in Chlamydia species from goats from Vietnam. Although, this is an interesting points which we will discuss in a next article including a larger number of samples. We are also working on standardizations of ELISA that is cheaper than PCR, of course.
Introduction:
Line 73-77
The statement describe results and conclusion which is not appropriate in the introduction section.
instead, should clearly state the objectives of the study.
Results: We agree with the Reviewer and modified the objectives of the study as follows:
“Since goats play a significant role in transmission of the disease to humans and domestic ruminant farming is common in Vietnam, aim of this preliminary study was to investigate the presence of Chlamydia 16SrRNA species in goats sampled in six small rural communities of Central Vietnam”
Results:
If samples were collected from different family owned small herds, could you indicate how many herds were positive? i.e were the 11 positive samples from 11 different herds or less?
Response: In result section and in table 1 these information have been added and now positive herds have been listed.
Line 90 Table 1: provide more descriptive table title
Response: We agree with your comment and the title has been modified as follows:
“BLASTn identities of the 16S rRNA sequence types obtained from the eleven infected goats clustered by farms”
table1: the authors indicated 60 samples were collected, why there are labels for goat 66 & 67 under
host column, also 7a & 8°
Response: We agree with your comment, we mantained the original number assigned by the vet that conduct the sampling. Since it could generate confusion, the list of infected samples was modified according to the herds from which they were collected as shown in Table 1.
2.2. Sequence analyses of the 16S rRNA positive samples
Line 92: The authors indicated a figure 1 in the first revision, but did not provide any figures. Please
explain?
Response: The first version of the paper wrongly indicated a figure of phylogenesis of Chlamydiaceae in goats, that we did not realize that and now the reference has been cancelled
Discussion
Lines 112-114:
If there were no previous reports on the prevalence of Chlamydia spp. in ruminants in Vietnam, how do know the impact of the disease on public health has largely been underestimated?
Response: Infact we agree with your comment and the sentence has been reformulated as follows:
“However, to the best of our knowledge, the prevalence of Chlamydia spp. in ruminants has never been reported in Vietnam and considering the high number of goats positive for Chlamydia spp. farmed in the six local human community, we speculate that the impact of the disease on public health is not to be underestimated.”
Lines 119-120: What do you mean by clinical samples? you already mentioned all animals were
asymptomatic.
Response: The sentence has been modified .
Materials and Methods
Line 185: over 6 months. Goats had no observable diseases, nor were they were injured
This statement was already mentioned at line 17
Response: This paragraph has been modified and now reads:
“All goats were adult (over 6 months) with average weight of about 15 kg. There was not previous history of abortion in the herds, even if chlamydiosis was suspected in the rural communities. Goats had no observable diseases or injuries. Moreover, sampled animals were all treated against ticks at the beginning of tick season. As goats in Vietnam are farmed for meat in small groups by families, a consent from the owners of the goats was taken before collection of blood samples used for DNA extraction and detection of the presence of Chlamydia spp. in the samples.”